



# Seismic radiation from wind turbines: observations and analytical modeling of frequency-dependent amplitude decays

Fabian Limberger[1,2], Michael Lindenfeld[1], Hagen Deckert[2] and Georg Rümpker[1]

[1]Institute of Geosciences, Goethe-University Frankfurt, 60438 Frankfurt am Main, Germany
[2]Institute for Geothermal Resource Management (igem), 55411 Bingen, Germany

*Correspondence to:* Fabian Limberger (limberger@igem-energie.de)

**Abstract**

In this study, we determine spectral characteristics and amplitude decays of wind turbine induced seismic signals in the far field of a wind farm (WF) close to Uettingen/Germany. Average power spectral densities (PSD) are calculated from 10 min time segments extracted from (up to) 6-months of continuous recordings at 19 seismic stations, positioned along an 8 km profile starting from the WF. We identify 7 distinct PSD peaks in the frequency range between 1 Hz and 8 Hz that can be observed to at least 4 km distance; lower-frequency peaks are detectable up to the end of the profile. At distances between 300 m and 4 km the PSD amplitude decay can be described by a power law with exponent $b$. The measured $b$-values exhibit a linear frequency dependence and range from $b = 0.39$ at 1.14 Hz to $b = 3.93$ at 7.6 Hz. In a second step, the seismic radiation and amplitude decays are modeled using an analytical approach which approximates the surface-wave field. Since we observe temporally varying phase differences between seismograms recorded directly at the base of the individual wind turbines (WTs), source-signal phase information is included in the modeling approach. We show that phase differences between source signals have significant effects on the seismic radiation pattern and amplitude decays. Therefore, we develop a phase-shift-elimination-method to handle the challenge of choosing representative source characteristics as an input for the modeling. To optimize the fitting of modeled and observed amplitude decay curves, we perform a grid search to constrain the two model parameters, i.e., the seismic shear wave velocity and quality factor. The comparison of modeled and observed amplitude decays for the 7 prominent frequencies shows very good agreement and allows to constrain shear velocities and quality factors for a two-layer model of the subsurface. The approach is generalized to predict amplitude decays and radiation patterns for WFs of arbitrary geometry.

## 1 Introduction

In recent years, debates on the emission of seismic waves produced by WTs and its potential effects on the quality of seismological recordings have led to increased research efforts on this topic. Main objectives are the characterization of WT-




induced seismic signals, the definition of protection radii around seismological stations and moreover the modeling-based
prediction of WT effects on seismological recordings in advance of the installation of WTs. Styles et al. (2005) reported about
discrete frequency peaks in seismic noise spectra that increase with wind speed and the rotation rate of a nearby WT and
assigned the observed peaks to vibration modes of the WT tower and rotor rotation. Zieger and Ritter (2018) as well as
Stammler and Ceranna (2016) confirmed discrete frequency peaks between 1 and 10 Hz and analyzed signal amplitude decays
with distance to the WTs described by a power law. Saccorotti et al. (2011) observed seismic signals with a frequency of about
1.7 Hz which were associated to WTs at distances of up to 11 km. Friedrich et al. (2018) used a migration analysis to identify
seismic signals from nearby WFs and were able distinguish between various WFs based on differences in frequency content.
Polarization analyses was used by Westwood and Styles (2017) to show that Rayleigh waves dominate the wave field emitted
from WTs. This observation was confirmed by numerical simulations (Gortsas et al., 2017). The increase of the noise amplitude
with the square root of the number of WTs ($\sqrt{N}$) was observed by Neuffer et al. (2019) based on WT shut-down tests. Lerbs
et al. (2020) proposed an approach to define protection radii, for example of 3.7 km around the Collm Observatory (CLL) in
Germany, using a power law to describe the spatial wave attenuation. Furthermore, the ground motion polarization near a
single WT was analyzed and provided insights to the interaction of WT nacelle movement and emitted seismic signals.

Approaches to model the seismic radiation from WTs are rare and focus mostly on modeling the ground vibration of a single
WT (Gortsas et al., 2017) or its operational components only (e.g. Zieger and Nagel, 2020), but not on wave field propagation
considering superimposed wave fields and amplitude decay with distance to multiple WTs simultaneously. However,
Saccorotti et al. (2011) used an analytical approach to model the observed amplitude decays by summing up the calculated
noise amplitudes produced by several WTs, including an intrinsic attenuation law, but did not study possible effects of multiple
WTs on the interference of the emitted wave fields.

In this paper, we present an analytical approach to model frequency-dependent seismic radiation and amplitude decays with
distance in comparison to robust long-term observed decay curves, measured at a WF in Uettingen (Bavaria). In a first step,
we derive distance-dependent noise spectra from recordings of up to six months duration and characterize the relation between
signal frequency and amplitude decay. We face the challenge of handling phase differences between multiple source signals
that have strong effects on the seismic radiation, due to significant changes in interference pattern of the superimposed wave
fields. We apply the phase-shift-elimination-method (PSE-method) to generate representative source signals as an input for
the analytical modeling of the observed amplitude decays. The comparison between modeled and observed amplitude decays
also allows to constrain the parameters of a simple 2-layer model of the subsurface. We further show to generalize the approach
to predict radiation patterns for arbitrary WF geometries.

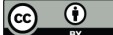
## 2 Observational data

Our surveys were conducted in the neighborhood of a WF in Uettingen, close to the city of Würzburg in Bavaria. The WF

consists of three WTs positioned in a NW-SE line with a spacing of 350 m resp. 450 m. The Nordex N117 type WTs have

2400 kW rated power and a tower height of 141 m. Maximum rotation rate is about 12 rpm (revolutions per minute). To

measure the amplitude decay of the seismic WT signals we deployed 19 seismic stations along a profile of 8.3 km length,

starting at the easternmost of the 3 WTs and running in NE direction which is approximately perpendicular to the geometrical

layout of the WF (Fig. 1). Additionally, we placed three stations in the WT basements in order to record the seismic source

signal of each WT. The instruments were installed between July and November 2019 and data recording will extent until

August 2021. All stations are equipped with Trillium Compact posthole sensors (20 s) and Centaur data loggers (Nanometrics)

recording continuously at a sampling frequency of 200 Hz. To improve the signal/noise conditions the sensors of the profile

stations were placed in shallow boreholes of 1 – 2 m depths.

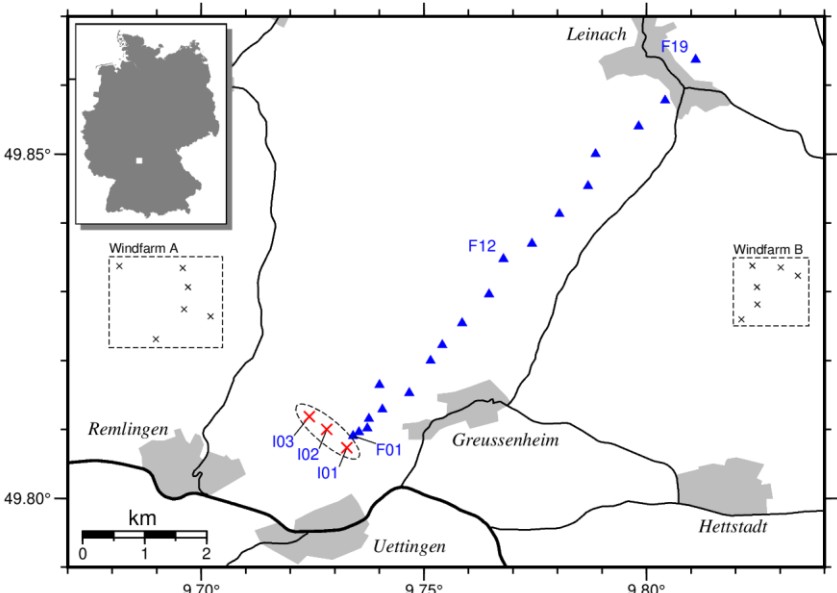

**Figure 1:** Location of the wind farm in Uettingen (red crosses) and seismic profile stations F01 to F19 (blue triangles). Three

additional seismic stations are positioned in the WT cellars (I01, I02, I03). Wind farms A and B (dashed boxes) are not targeted

by our experiment but are located in the area.

### 2.1 Calculation of average PSDs

We analyzed a continuous dataset between September 2019 and March 2020, covering a range of 159 to 207 days depending

on the exact station installation date. We associate the measured amplitudes in the seismic waveform data with the




corresponding WT parameter (in this case "rotor speed") at a resolution of 10 minutes. For this reason, the recordings of each profile station were split into 10-minute segments which were transformed to power spectral density spectra using the method of Welch (1967). Each of these spectra was then sorted according to the respective rotor speed into bins of 1 rpm width. With this procedure we generated close to 10000 single PSDs within the bin of maximal rotor speed (11 – 12 rpm) called "full

power" status for each station, and about 2000 single PSDs for the "zero power" status of the WT (0 – 1 rpm). In order to reliably remove outliers and reduce the impact of local transient noise (e.g., traffic on nearby roads), we used the 25% quantile of the single PSDs in each bin to calculate the final average PSD spectra.

Figures 2 and 3 show the resulting average PSDs for the "full power" and the "zero power" WT status, respectively. Besides the strong microseismic peak at about 0.2 Hz, we identified 9 peaks of significant energy centered at 1.14, 1.7, 2.3, 3.5, 4.8,

6.0, 7.6, 10.5, and 17.2 Hz, respectively. All of them show a systematic amplitude decrease with increasing station distance, indicating that their origin is located at the WT. For peaks 1 to 7 we fitted the observed amplitude decay with a power law model (see next section). Because of the rapid amplitude decay at frequencies > 10 Hz we were not able to reliably fit the peaks 8 and 9. For comparison we show the respective average PSDs recorded during "zero power" status (0 – 1 rpm) in Fig. 3. In this case the observed spectral peaks 1 to 9 have completely disappeared. The remaining (sharp) peaks show no systematic

dependence which is an indication that their origin is not related to the WTs.

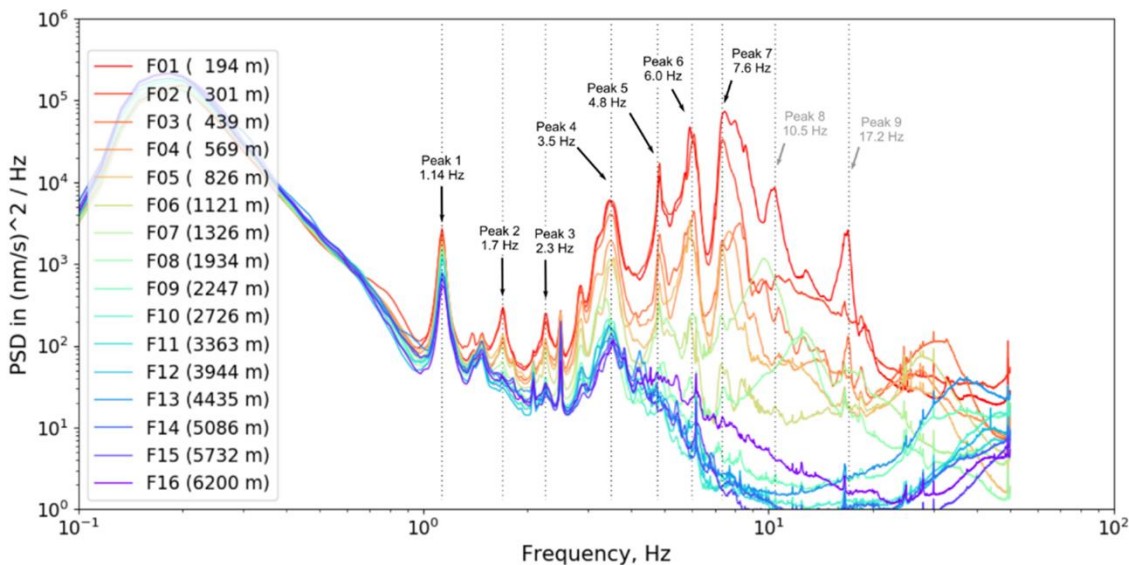

**Figure 2:** Average PSD spectra at "full power" status (11-12 rpm), calculated at profile stations F01 to F16 in the time range from September 2019 to March 2020. The distance of each station to the WT is colour coded and indicated in the figure legend.

In total 9 energy peaks are identified between 1.14 Hz and 17.2 Hz, which show a systematic amplitude decrease with increasing station distance. The amplitude decays of peaks 1 to 7 have been measured and fitted by a power law.





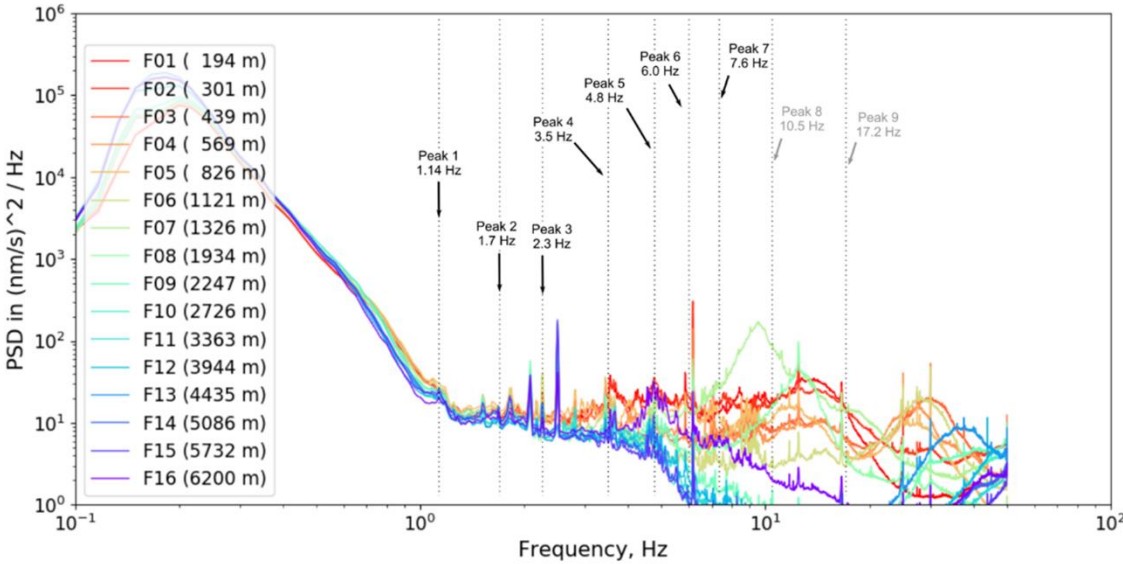

**Figure 3:** Average PSD spectra at "zero power" status (0 - 1 rpm), calculated at profile stations F01 to F16 in the time range from September 2019 to March 2020. The identified peaks at "full power" (Fig. 2) have disappeared. The remaining sharp peaks show no systematic decrease with increasing distance, indicating that they have a different origin.

## 2.2 Power-law fitting of the observed amplitude decay

To quantify the amplitude decay, the respective peak maxima of the "full power" PSDs (Fig. 2) were picked at each station. Figure 4 shows the resulting attenuation curves for peak 1 (1.14 Hz) to peak 7 (7.6 Hz) using a double-logarithmic representation, i.e. the logarithm of peak amplitude is shown versus the logarithm of the station distance. If the amplitude decay corresponds to a power law, which is the basic assumption, there should be a linear correlation between log(amplitude) and log(distance). The attenuation factor, $b$, can then be calculated as the slope of a linear fit of the attenuation curves. As Fig. 4 shows, the measured amplitude decays can be described in good approximation with a power law between station F02 and F12, which corresponds to a distance range of 300 m to 4000 m. Beyond F12 the measured amplitudes are increasing with larger distances, except for peak 1 (1.14 Hz), and it was not possible to identify clear peak maxima, since the background noise dominates the spectra. Towards the end of the seismic profile the stations are getting closer to a high-speed railway track and populated areas with raised ambient noise conditions, which might explain the observed excessive amplitudes in this region. However, also the first two stations of the profile - F01 (194 m) and F01a (239 m) - show deviations from a power law attenuation. Due to the proximity of these stations to the WT, the amplitudes may also be affected by near-field effects.



**Figure 4:** Double logarithmic representation of PSD amplitude decay at seven different peak frequencies. Blue circles mark the measured amplitudes from station F01 (194 m) to station F19 (8413 m) at "full power" status of the WT. Filled symbols denote data points that were used for power law fitting (red lines) between station F02 (301 m) and F12 (3944 m) with attenuation factor $b$ and correlation coefficient $R^2$.

For these reasons we decided to restrict the analysis of the amplitude decay to the distance range between 300 m and 4000 m and to estimate the attenuation factor, $b$, by a linear least squares fit between station F02 and F12 for all of the seven peak frequencies. The results show a systematic increase with frequency and yield values from $b = 0.39$ at 1.14 Hz up to $b = 3.93$



at 7.6 Hz. In Fig. 5 we show the frequency dependence of the attenuation factor *b*. It exhibits a nearly perfect linear relationship
between *b* and frequency, at least within the analyzed frequency range from 1.14 Hz to 7.6 Hz. The comparison to results from other authors is discussed below (section 5).

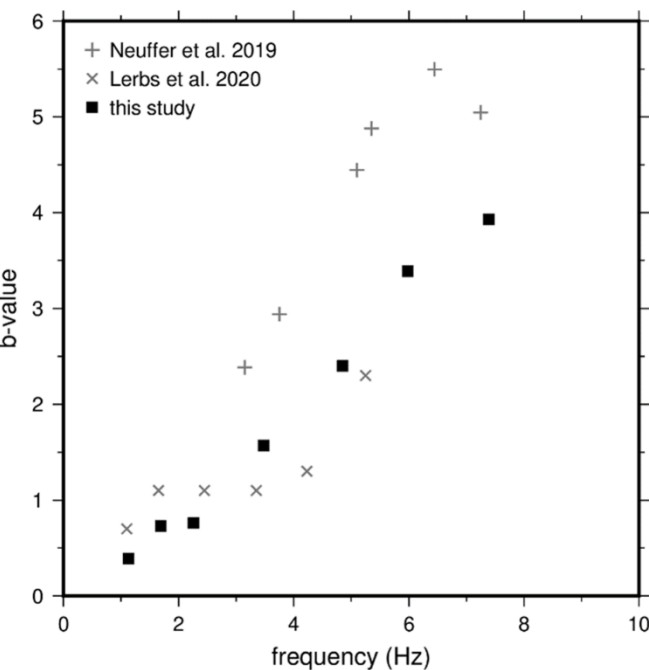

**Figure 5:** Frequency dependence of *b*-values for peak 01 to peak 07 (filled symbols, cf. Fig. 4). Plus signs and crosses mark calculated *b*-values of Neuffer et al. (2019) and Lerbs et al. (2020), respectively.

**2.3 Observation of phase shifts between multiple WT vibrations**

Each WT can be considered as a seismic source. By analyzing the seismograms measured simultaneously in the three WTs (I01, I02 and I03) of the WF Uettingen, we observe phase shifts between the individual wave forms (Fig. 6a). As an example, three time series (vertical component), each recorded in one of the WT cellars during a rotation rate of about 11.5 rpm, are filtered to a narrow bandwidth around frequency peak 1 with 1.14Hz (1.10 – 1.18 Hz) and are compared within a time window
of 22 s. In the first 2 s, the signal phase of seismic station I03 is shifted by $\pi$ compared to signal I01 and I02, which are in phase. Between 10 and 13 seconds, all signals are almost in phase, which consequently means that the WTs are vertically vibrating in phase. After 15 seconds, all three signals are shifted to each other and are not in phase anymore. For a longer time period of 1 hour, the phase shift between signals measured at I01 and I02 are determined using a cross correlation analysis with a moving time window of 5 s (=720 time segments) along the 1 hour time segment. The temporal shift is converted to the
corresponding phase shift between $-\pi$ and $\pi$ for each window. The distribution of all 720 resulting phase shifts is almost





uniform (Fig. 6b) and shows no systematic behavior with time (Fig. 6c), which leads to the conclusion that phase differences between source signals appear rather randomly, especially over longer time periods.

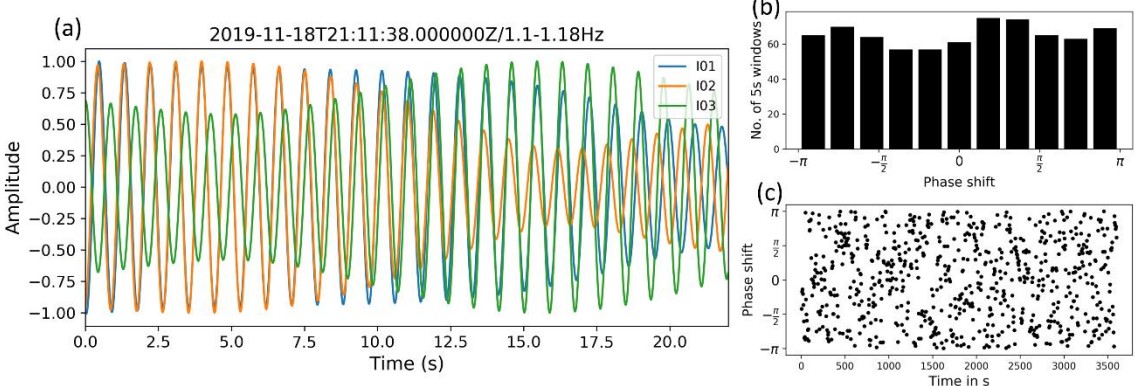

**Figure 6:** (a) Comparison of seismograms (vertical components) measured simultaneously in each of the three WTs at rotation rate of 11-12 rpm. Waveforms are filtered to 1.10-1.18 Hz and amplitudes are normalized to their maximum. (b) Distribution of phase shift between signals in 5 s time segments measured in two WTs (I01 and I02) during a period of 1 hour with WT rotation rates of 11-12 rpm. (c) To (b) corresponding phase shifts with time.

## 3 Analytical modeling approach

In the following section, we model the observed amplitude decays and set up a mathematical formulation that includes a source function, attenuation factors, geological properties and the superposition of multiple wave fields (produced by multiple WTs). In view of the observation that the source signals of neighboring WTs are not in phase, we study the influence of possible signal phase differences on the amplitude decay and propose a solution how to account for or "eliminate" this effect in the calculation.

### 3.1 Surface wavefield approximation

Previous research suggests that mainly vertically polarized Rayleigh waves are emitted from WTs and dominate the WT-induced seismic noise (Westwood and Styles, 2017; Neuffer and Kremers, 2017; Gortsas et al., 2017). In our models, we assume surface-wave amplitudes to decay proportionally to $r^{-1/2}$ (with distance $r$ to the source) due to geometrical spreading of the surface wave front on a cylindrical area in the 2D surface plane

$$G = \sqrt{\frac{r_0}{r}}, \tag{1}$$





where $r_0$ is a reference (minimal) distance (Bugeja, 2011).

Geometrical spreading is independent of wave frequency. In addition, attenuation due to intrinsic absorption reduces the wave amplitude with distance to its source

$$D = \exp^{\frac{-wr}{2cQ}} \quad .$$ (2)

The damping factor, $D$, depends on frequency $w = 2\pi f$, seismic wave velocity $c$ and again the travel distance $r$ of the wave

(Bugeja, 2011). Furthermore, D is a function of the seismic quality factor $Q$, which describes the loss of energy per seismic wave cycle due to anelastic processes or friction inside the rock during the wave propagation. The damping of the wave is decreasing with increasing Q. The source signal $S(t)$ itself is approximated by a continuous periodic cosine-function to simulate the periodic motion at the base of the WT in vertical direction

$$S(t) = A \cos(kr - wt + \Phi) \quad ,$$ (3)

where $S(t)$ is a function of time $t$, signal frequency $w = 2\pi f$, the amplitude calibration factor $A$, wave number $k = w/c$ and signal phase $\Phi$.

Assuming a homogeneous half-space, the wave amplitude can be calculated for any distance $r$ to the source (Fig. 7). Considering $N$ source points (WTs), the amplitude at each point and hence the total wave field is derived by summation over all $N$ wave fields

$$Z(t) = \sum_{i=1}^{N}[\, S_i(t)\, G_i\, D_i] = \sum_{i=1}^{N}[\, A_i \cos(k_i r_i - w_i t + \Phi_i) \cdot \sqrt{\frac{r_{0,i}}{r_i}} \cdot \exp^{\frac{-w_i r_i}{2c_S Q_S}}\,] \quad ,$$ (4)

where index $i$ corresponds to source point $i$ and the relative radial distance to the source points is given by $r_{0,i}/r_i$. Shear quality factor $Q_S$ and seismic shear wave velocity $c_S$ are model parameters and define the properties of the material the wave is traveling through. $Z(t)$ is the superposition of the individual wave fields and can be calculated at any time $t$.



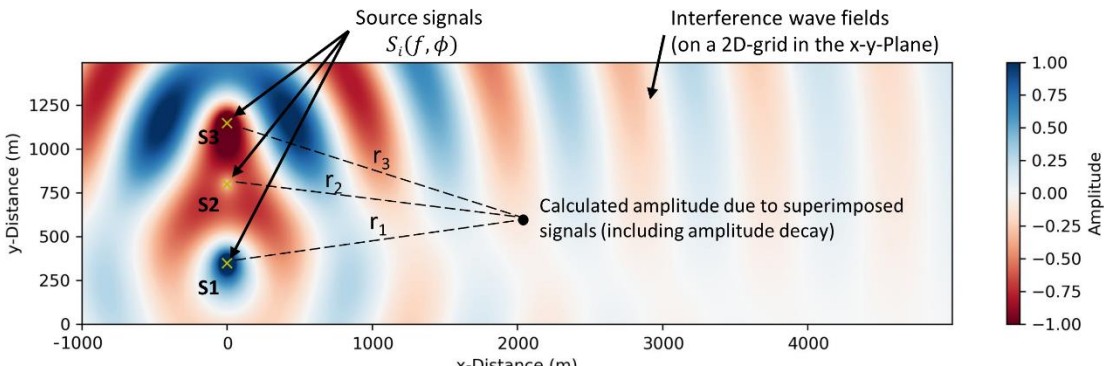

**Figure 7:** Schematic figure of the analytical modeling approach. Amplitudes as functions of time are calculated at points (x,y).

By modeling the interference wave field through time, this approach allows to derive root mean square amplitudes (RMS amplitudes) at any point at the surface. Here, the amplitude calibration factor $A_i$ will be set to 1 for every source signal, since all three WTs of the Uettingen WF are of the same type. It should be noted that body waves (P, S) are not directly considered in this modeling approach. However, the Rayleigh wave velocity $c_R$ is generally slightly lower than the shear wave velocity $c_S$, whereas the $c_R/c_S$ ratio depends on the Poisson ratio $\nu$ (e.g. Rahman and Michelitsch, 2006). Assuming theoretical values of $\nu$ from 0.0 to 0.5, the ratio $c_R/c_S$ reaches values between 0.87 and 0.95 (Leiber, 2003; Hayashi, 2008). However, it is possible to approximate surface wave fields using shear wave velocity (Kumagai et al., 2020). Assuming, that mainly Rayleigh waves are radiated by WTs, the penetration depth of Rayleigh waves, which is influenced by the physical properties of near-surface geological layers and the wavelength, plays an important role in approaching the modeling of WT-induced seismic wave propagation. The penetration depth of Rayleigh waves is widely studied, however, so far there is no general consensus on the penetration depth in relation to the seismic wavelength $\lambda$. Based on results of Hayashi (2008), Kumagai et al. (2020) claim that surface wave velocity reflects the average S-wave velocity in a depth between $\frac{1}{4}\lambda$ and $\frac{1}{2}\lambda$, whereas $\frac{1}{3}\lambda$ is often chosen to be the most suitable assumption (e.g. Larose, 2005). Moreover, it is common to derive depth information from observed wave attenuation applying modeling or tomography methods to seismological data (e.g. Siena et al., 2014). Due to Rayleigh wave dispersion, it is known that low frequency surface waves reach deeper into the subsurface, thus travelling through materials with likely higher $Q$ and seismic velocities $c$. Consequently, the damping is reduced compared to high frequency surface waves (Karatzetzou et al., 2014; Farrugia et al., 2015). Taking this into account, we use the following relation for wavelength depth conversion

$$d_{\lambda/3} = \frac{1}{3}\lambda \qquad (\lambda = \frac{c_S}{f}) \qquad . \qquad\qquad (5)$$





In this study, we take advantage of the link between frequency-dependent amplitude decays (depending on $Q_S$ and $c_S$, Eq. 4) and surface wave penetration depth to derive information about shear wave velocities and quality factors in the subsurface (Eq. 5).

**3.2 Effect of source-signal phase on seismic radiation and amplitude decay**

Since we observe significant changes of the phase shifts between signals measured at the three WTs (section 2.3), we aim to
study its effect on the wave field that is emitted by the three WTs in Uettingen. Hence, three wave fields are calculated using three different source phase compositions assuming a 1.14 Hz source signal frequency, 1500 m/s wave velocity and a quality factor of 30 as an exemplary model. Source points are located at $x_1 = x_2 = x_3 = 0$ m and $y_1 = 350$ m, $y_2 = 800$ m and $y_3 = 1150$ m (Fig. 7). Amplitudes are calculated along a profile extending from source S1 and perpendicular to the WF line, which approximates the real geometry of the WF and seismic profile in Uettingen. The results show a clear dependence of the
amplitude decays on the source signal phase composition (Fig. 8). In addition, amplitudes at the end of the 5000 m long profile differ significantly to each other. In the third scenario (Fig. 8c) the expected amplitude is a quarter of the amplitude that is reached if the WTs are vibrating in phase (Fig. 8a). Furthermore, strong effects appear in the first 2 km of the profile. Scenario (a) shows increased amplitudes due to constructive wave interference in the near field, whereas scenario (c) indicates a rapid decay of amplitudes within the first 1000 m of the profile. Scenario (b) shows a smoother and steadier decay of the amplitudes
and reaches an amplitude at the end of the profile that is reduced by a fifth compared to scenario (a). These exemplary scenarios demonstrate only three out of infinite possibilities of different source signal phase compositions. Taking this into account, the seismic radiation of a WF is affected by phase differences of the source signals which can lead to strong changes in the wave field interference.

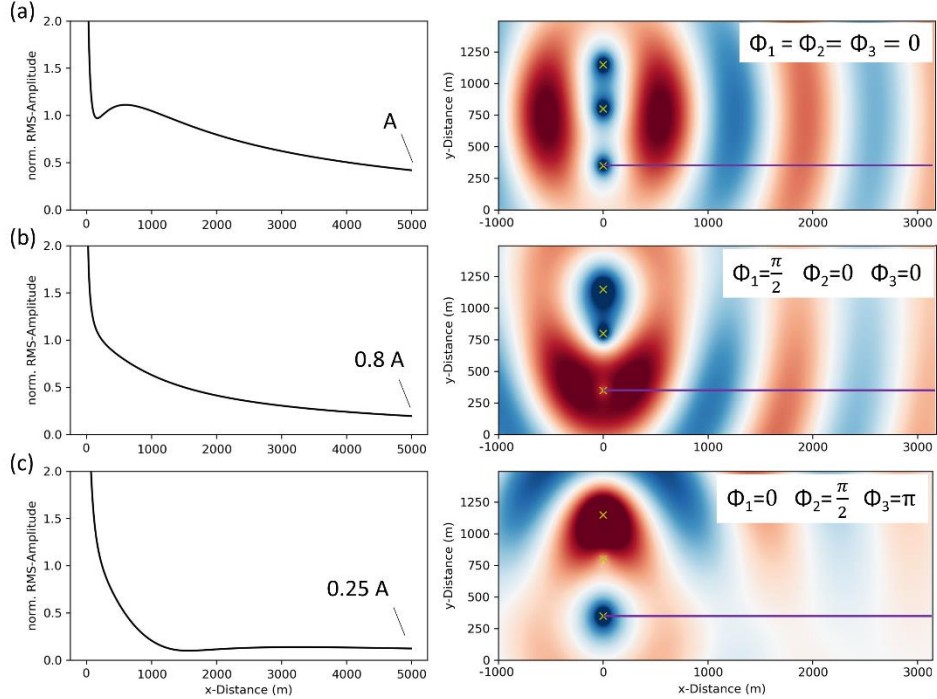

**Figure 8:** Calculated amplitude decay curves (in direction of the magenta line) for three scenarios with different source signal phase compositions using (a) $\Phi_1 = \Phi_2 = \Phi_3 = 0$ (vibration in phase), (b) $\Phi_1 = \pi/2$, $\Phi_2 = 0$, $\Phi_3 = 0$ and (c) $\Phi_1 = 0$, $\Phi_2 = \pi/2$, $\Phi_3 = \pi$. Index 1 represents the source point S1. All decay curves are normalized to the amplitude at $x = 300$ m.

### 3.3 Phase-shift-elimination and data fitting

In this section we propose a method how to handle the observed time-varying source signal phases and its effect on the seismic radiation using the assumption of a random appearance of signal phase constellation of multiple WTs, especially regarding long time periods. To define representative source signals, we developed a phase-shift-elimination-method (PSE-method). Within this PSE-method, 500 radiation patterns (= wave fields) are calculated using random signal phases $\Phi$ (between 0 and $\pi$) for each individual source signal. All calculated 500 wave fields are then averaged and the average amplitude decay is extracted along the profile, which in turn is independent of the individual source signal phases. We experienced, that the wave field averaging process and hence final amplitude decay calculation is sufficiently stable after 500 wave field simulations, whereas a number of < 100 seems too low to generate a reproducible result. We apply this method to the Uettingen WF set up and compare the modeling results with the observed amplitude decays in Uettingen during rotations rates between 11 rpm and 12 rpm (all WTs under "full power"). Since the PSD-values are proportional to squared ground motion amplitudes, we compare our modeling results with the square root of the observed PSD-amplitudes. The analysis is performed for signals with center frequencies of 1.14 Hz, 1.69 Hz, 2.26 Hz, 3.5 Hz, 4.85 Hz, 5.98 Hz, and 7.6 Hz, representing the 7 PSD peaks in Uettingen



(see section 2.1). For comparison, all decay curves are normalized to the amplitude measured in 300 m distance (seismic station F02), to be consistent with the attenuation analysis presented in section 2.2. The calculated radiation pattern covers an area of 6000 in length (x) and 1500 m in width (y) with a grid space of 10 m.

Calculated and observed data are fitted by a $Q_S$-$c_S$-grid search to find the best model parameters. The data is grouped into
signals with low frequencies < 4 Hz (1.14 Hz, 1.69 Hz, 2.26 Hz, 3.5 Hz) and high frequencies > 4 Hz (4.85 Hz, 5.98 Hz, 7.6 Hz) to distinguish between shallow and deep geological effects on the amplitude decay due to frequency-dependent penetrating depth of surface waves. All amplitude decays per group are fitted with one $Q_S$-$c_S$-model. To set up the grid search, model parameter $c_S$ are varied from 400 to 3000 m/s using steps of 20 m/s and parameter $Q_S$ are varied between 6 to 250 using a step size of 2. An averaged (500 decay curves) amplitude decay with distance is calculated for each combination of $Q_S$ and $c_S$ and
is compared to the observed data by calculating the root-mean-square error

$$RSME = \sqrt{\frac{\sum_{i=1}^{M}(obs_i - sim_i)^2}{M}} \tag{6}$$

where $M$ represents the 14 seismic stations along the profile that are included in the fitting process. This process is performed for 16114 different models per frequency (Fig. 9).

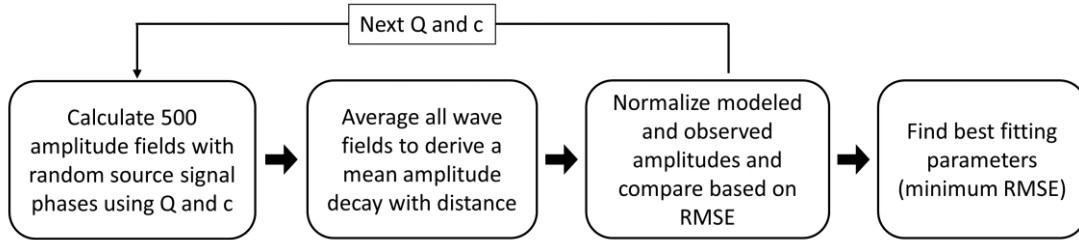

**Figure 9:** Description of the fitting process to find best model parameters from the comparison of calculated and observed amplitudes.

Moreover, the normalized root-mean-square error (NRMSE) is obtained to quantify the fitting quality. To determine the NRMSE, each RMSE is divided by the range (maximum value - minimum value) of the observation amplitudes for each frequency in order to scale the comparison between the data sets. The total NRMSE is then given by the mean of all normalized
RMSE

$$NRMSE = \frac{RMSE}{obs_{max} - obs_{min}} \quad . \tag{7}$$





# 4 Results

To fit modeled and observed amplitudes we performed a separate grid-search for both the group with high frequency and with low frequency signals. During each individual fitting process, the PSE-method was applied to ensure results that are

independent of source signals phases. Regarding the group of low frequency signals (<4Hz), we obtain $Q_S = 40$ and $c_S = 960$ m/s (Tab. 1) as the best model parameters. The values of the 20 best models range between 36 and 60 for $Q_S$ and 920 m/s and 1040 m/s for $c_S$. Regarding the group of high frequency signals (>4Hz), we obtain $Q_S = 16$ and $c_S = 540$ m/s as the best parameters. Results of the 20 best models range between 12 and 32 for $Q_s$ and between 540 and 660 m/s for $c_S$ (Fig. 12). By fitting two frequency groups, we can derive a two-layer model, after converting the frequency-dependent wavelength to the

corresponding penetration depth (Eq. 5). Thus, we expect a shear wave velocity of 540 m/s until 37 m depth and 960 m/s until 280 m depth (Fig. 13). However, transition between the two layers (37 m to 91 m) is not clearly defined due to missing information for frequencies between 3.5 Hz and 4.85 Hz. The velocity error is approximated by the range of the 20 best models (Fig. 12). During the fitting process, we noticed that it was not possible to fit all seven amplitude decays with only one $c_S$-$Q_S$-model successfully, especially regarding signals with frequencies > 4 Hz. A homogeneous model is consequently not

reasonable in this case. However, the corresponding results are given in the appendix (Fig. A1). Modeled and observed data are generally in very good agreement for each of the seven analyzed frequencies (Fig. 10 and 11). The very slow decrease of observed amplitudes, especially at 1.14 Hz (Fig. 10a), and the relatively strong decrease of signals with 7.6 Hz (Fig. 11c) are simulated correctly and confirm a higher attenuation with higher frequencies, as expected. For a frequency of 1.14 Hz, between $x = 2000$ m and 4000 m, modeled amplitudes are underestimated in comparison to the observations. Minor deviations

between modeled and observed data for frequencies > 4 Hz might be explained by local effects that are not represented in our laterally homogeneous models. Interestingly, a local increase of amplitude with distance are observable in the real data, prominently for 3.5 Hz and 4.85 Hz signals, as well as in the simulated data (Fig. 10d and 11a). This undulation is likely caused by superimposed wave fields of multiple WTs, as indicated by the modeled radiation pattern. Moreover, the sensibility concerning the source signal phase compositions decreases clearly with increasing frequency, which indicates that effects of

phase differences between source signals are more significant for lower frequencies.

**Table 1:** Best model parameters ($c_S$ and $Q_S$) to fit observed and calculated amplitude decays of low and high frequency signals. Depth $d$ is estimated by assuming a surface wave penetration depth of $\lambda/3$.

|  | $f$ in Hz | $c_{S,\text{mean}}$ in m/s | $Q_S$ | $d_{\lambda/3}$ in m |
|---|---|---|---|---|
| Low frequency group | 1.14, 1.69, 2.26, 3.5 | 960 | 40 | 91-280 |
| High frequency group | 4.85, 5.98, 7.6 | 540 | 16 | 0-37 |





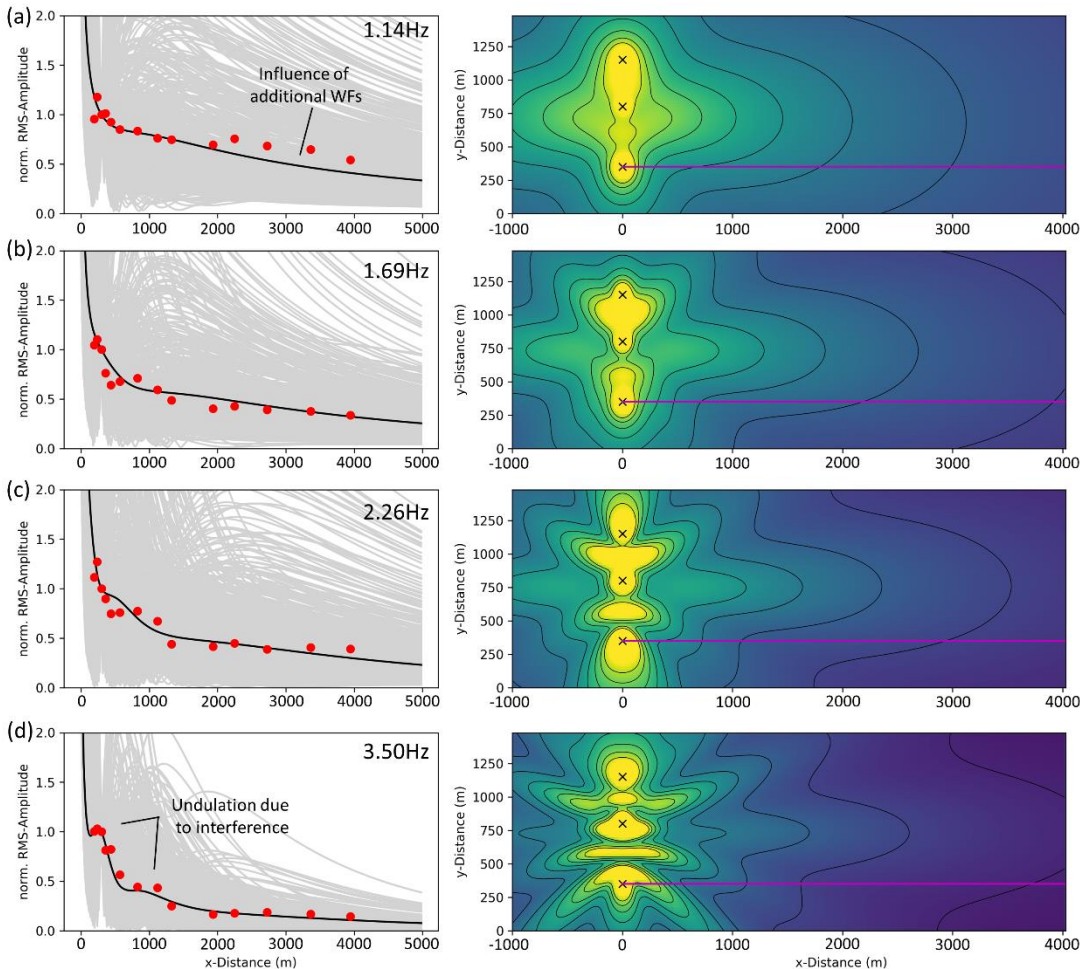

**Figure 10:** Averaged modeled radiation patterns (right) and averaged amplitude decays (left, black line) along the profile (magenta line) by averaging 500 wave fields and decay curves (gray lines), based on random $\phi$ (between $0,\pi$) to eliminate the effect of phase differences between source signals. Red dots represent the observed amplitudes in Uettingen at (a) 1.14 Hz, (b) 1.69 Hz, (c) 2.26 Hz and (d) 3.50 Hz.





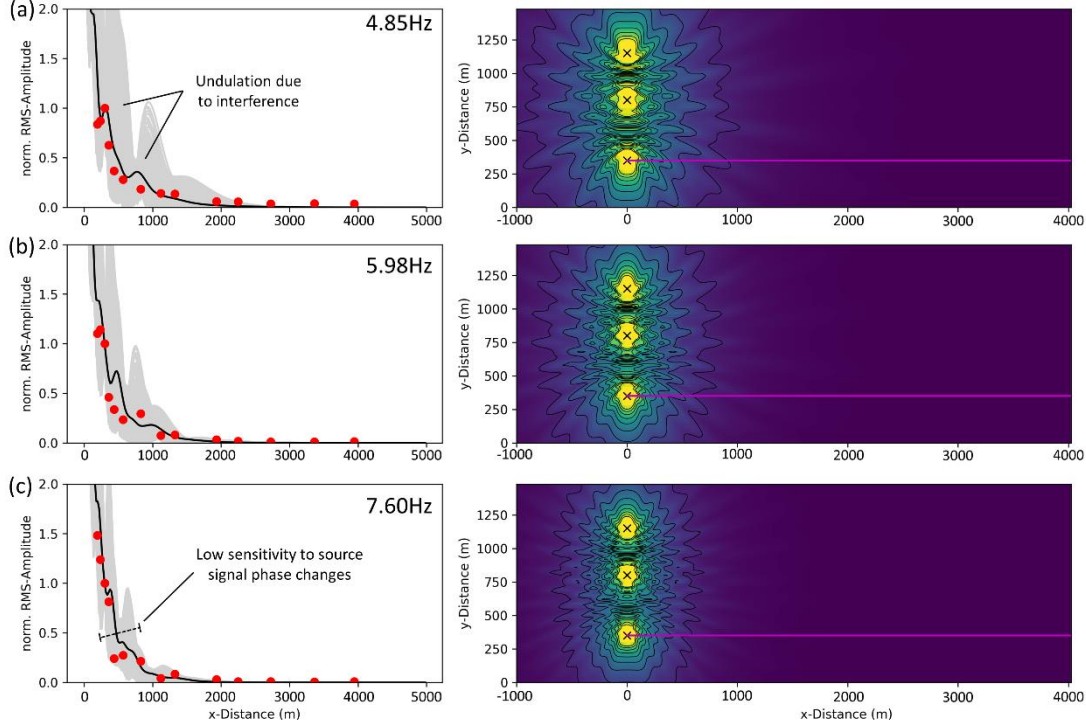


**Figure 11:** Averaged modeled radiation patters (right) and averaged amplitude decays (left, black line) along the profile (magenta line) by averaging 500 wave fields and decay curves (gray lines), based on random $\phi$ (between $0,\pi$) to eliminate the effect of phase differences between source signals. Red dots represent the observed amplitudes in Uettingen at (a) 4.85 Hz, (b) 5.98 Hz and (c) 7.60 Hz.





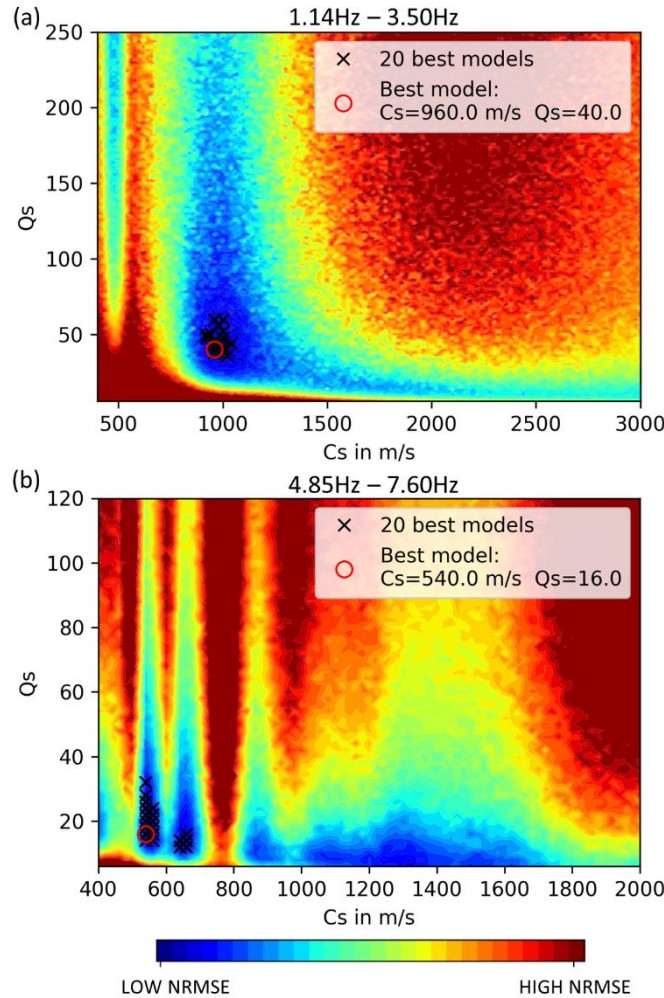


**Figure 12:** Distribution of the error (NRMS) of the fit between modeled and observed amplitude decays obtained by a $Q_S$-$c_S$-grid search. The 20 best models (black x) and the very best model (red circle) that fit amplitude decays of signals with (a) 1.14 Hz, 1.69 Hz, 2.26 Hz and 3.50 Hz and (b) 4.85 Hz, 5.98 Hz and 7.60 Hz.


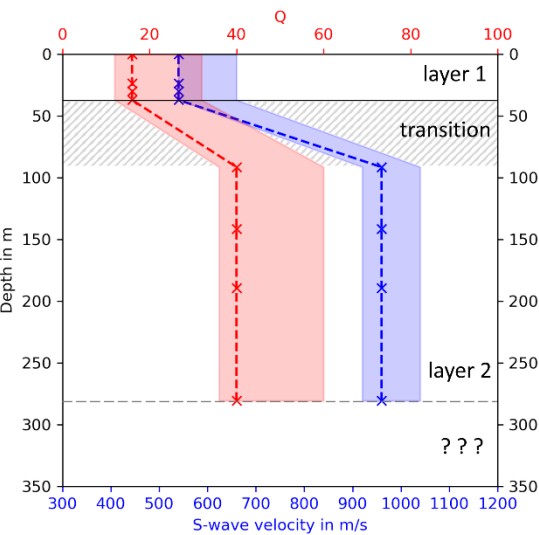

**Figure 13:** Two-layer model including $c_S$ and $Q_S$ information with depth.

**5 Discussion**

The aim of this study is to present reliable amplitude decays of seismological signals produced by multiple WTs and to model these amplitude decays with an analytical approach. The propagation of WT-induced seismic signals has been the subject of numerous studies. Many authors found that the amplitude decay with increasing distance ($r$) between WT and observation point can be described by a power law of the form $1/r^b$. In general, the absorption factor, $b$, increases with increasing frequency. Results found in our study show a near-perfect linear increase of the b-value with frequency and range from $b = 0.39$ at 1.14 Hz up to $b = 3.93$ at 7.6 Hz. The $b$-factors derived by the various authors cover a broad range of values, even for similar frequency ranges. Flores Estrella et al. (2017) published $b$-values from 0.73 to 1.87 for frequencies between 2.7 Hz and 4.5 Hz. Zieger and Ritter (2018) derived values from 0.78 to 0.85 at $1 - 4$ Hz, and $b = 1.59$ at 5.5 Hz. The results from Lerbs et al. (2020) range between 0.7 and 1.3 at $1 - 4$ Hz, and $b = 2.3$ at 5 Hz. Neuffer et al. (2019) derive $b$-values of 2.4 at 3 Hz and values of $b > 5$ at frequencies of $6 - 7$ Hz. In Fig. 5, we compare the $b$-values of Neuffer et al. (2019) and Lerbs et al. (2020) with our results. The studies of Neuffer et al. and Lerbs et al. yield a similar frequency dependence. However, the results of Neuffer et al. (2019) show systematically higher $b$-values. This observation could be due to different geological conditions with stronger attenuation effects during wave propagation. Furthermore, Neuffer et al. (2019) used so-called "differential" PSD spectra to measure the peak amplitude decay. These amplitudes are calculated from the difference between



the PSD peaks at "full-power" and the PSD peaks at "zero power" which could lead to an overestimation of the amplitude decay. Lerbs et al. (2020) get similar *b*-values, however, compared to our results the scatter is significantly larger. Most authors
explain the observed *b*-value scattering by different local geological conditions that influence the attenuation of the emitted seismic WT signals. It should be noted, however, that some of the above-mentioned studies use relatively short time windows to estimate the spectral amplitudes at increasing distances. Flores Estrella et al. (2017) analyze time series of 2 hours lengths, Lerbs et al. (2020) use 6 hours. In this case the measured amplitudes could be affected by transient signals such as earthquakes or local anthropogenic noise sources, which may result in uncertain *b*-value estimates. In contrast, Neuffer et al. (2019) extend
the analysis to 6.5 weeks. Since the knowledge of the amplitude decay plays a fundamental role in the modeling of the WT signals, we decided to use significantly longer time windows (6 months) in order to derive robust average PSD spectra at the installed profile stations.

In terms of modeling approaches, most of the recent publications focus on modeling the seismic signals that are emitted by one single WT (e.g. Gortsas et al., 2017), or the whole WF is considered as one emitting source. Since we observe time-varying
phase differences between the signals that are measured directly at the three individual WTs of the WF Uettingen, we propose that this effect must be included in the modeling of WFs. Our observations confirm the significance of phase differences between the seismic signals from the WTs of a wind farm and that the signal phase of a single WT is not stable over time. Hence, we expect that phase differences between source signals vary randomly, which was already presumed by Saccorotti et al. (2011). Superimposed wave fields lead to constructive and destructive interferences (which e.g. depend on signals phases)
and affect the spatial amplitude decay, as we can show in this study (Figure 8). Similar to our approach, Saccorotti et al. (2011) modeled amplitude decays on the basis of superimposed wave fields and attenuation laws but did not include phase-shift variations between signals of the WTs. However, they noticed that the increase of noise depends on WT number, which was later shown by Neuffer et al. (2019). Saccorotti et al. (2011) suggest that more accurate results can be derived by considering WTs that are not vibrating in phase. Here, we can prove the randomness of these phase differences between WTs and propose
a solution by applying the PSE-method to the modeling. Only with this consideration we can reproduce the observed amplitude decay. The PSE-method (averaging 500 wave fields calculated with random signal phases) is generally difficult to apply if full wave form propagation simulation is needed (e.g. FEM methods), since the required computation time would increase rapidly.

For the Uettingen WF the discrepancy between observed and simulated amplitude decays for 1.14 Hz in distances larger than 2000 m to the WTs are likely due to other nearby WFs. We assume that the low frequency signals of these WFs travel farther
compared to higher frequency signals and are measured in addition to the signals from the targeted three WTs in Uettingen (Fig. 1). Interestingly, the sensibility to source signal phases (gray lines in Fig. 10 and 11) is significantly higher for 1.14 Hz signals than 7.6 Hz and is generally decreasing with increasing frequency. This indicates that the signal phases are not that important for higher frequencies than for lower frequencies (e.g. 1.14 Hz). It should be noted that some of the individual input source signal phase compositions lead to decay curves that could not fit the observation data at all. This is solved using the
PSE-method.





Lerbs et al. (2020) proposed a solution which describes the wave attenuation with distance using an attenuation model solely based on a power law assumption (*b*-values). This approach does not allow a more universal application to other WFs or regions since *b*-values are not directly assigned to geological properties. The approach used in our study includes the intrinsic attenuation factor, which depends on two geological parameters, the seismic wave velocity and quality factor. A homogeneous
half-space is the basic assumption within our model. However, we show that the effect of layered media in the underground should be considered assuming frequency-dependent velocity and quality factors, due to significant dispersion effects of surface waves. It is generally an advantage to include geological properties in the model: 1) to consider actual physical properties of the medium the waves are traveling through and 2) to enable the possibilities of studying the effect of various geological conditions on the seismic radiation and amplitude decays.

To demonstrate the capability and possible application of the modeling approach used in this study, we modeled the radiation pattern of the original WF in Uettingen for 1.14 Hz and 7.6 Hz signals and compared the results with the case that three imaginary WTs are arbitrarily added to the existing WF (Fig. 14). Model parameters are $c_S$=960 m/s and $Q_S$=40 for 1.14 Hz and $c_S$=540 m/s and $Q_S$=16 for 7.6 Hz. The pattern of the radiation for 1.14 Hz signals is clearly affected by adding 3 WTs to the WF in Uettingen, whereby amplitudes are significantly increased, even in remote areas, for example in the NNW of the
WF (Fig. 14a and c). The effect on the characteristic radiation of 7.6 Hz signals is neglectable since the signal amplitude is damped rapidly in both cases, modeling 3 WTs and 6 WTs (Fig. 14b and c). As the demonstration shows, the modeling approach allows to estimate the characteristic seismic radiation pattern of an arbitrary WF in order to identify locations of low or high noise amplitudes or to evaluate WF geometry effects. Furthermore, the source locations, source signal frequencies and amplitudes as well as the expected local underground are free to choose (with limitations regarding its complexity), which
enables an approximation of the surface wavefield emitted by WFs with various layouts.

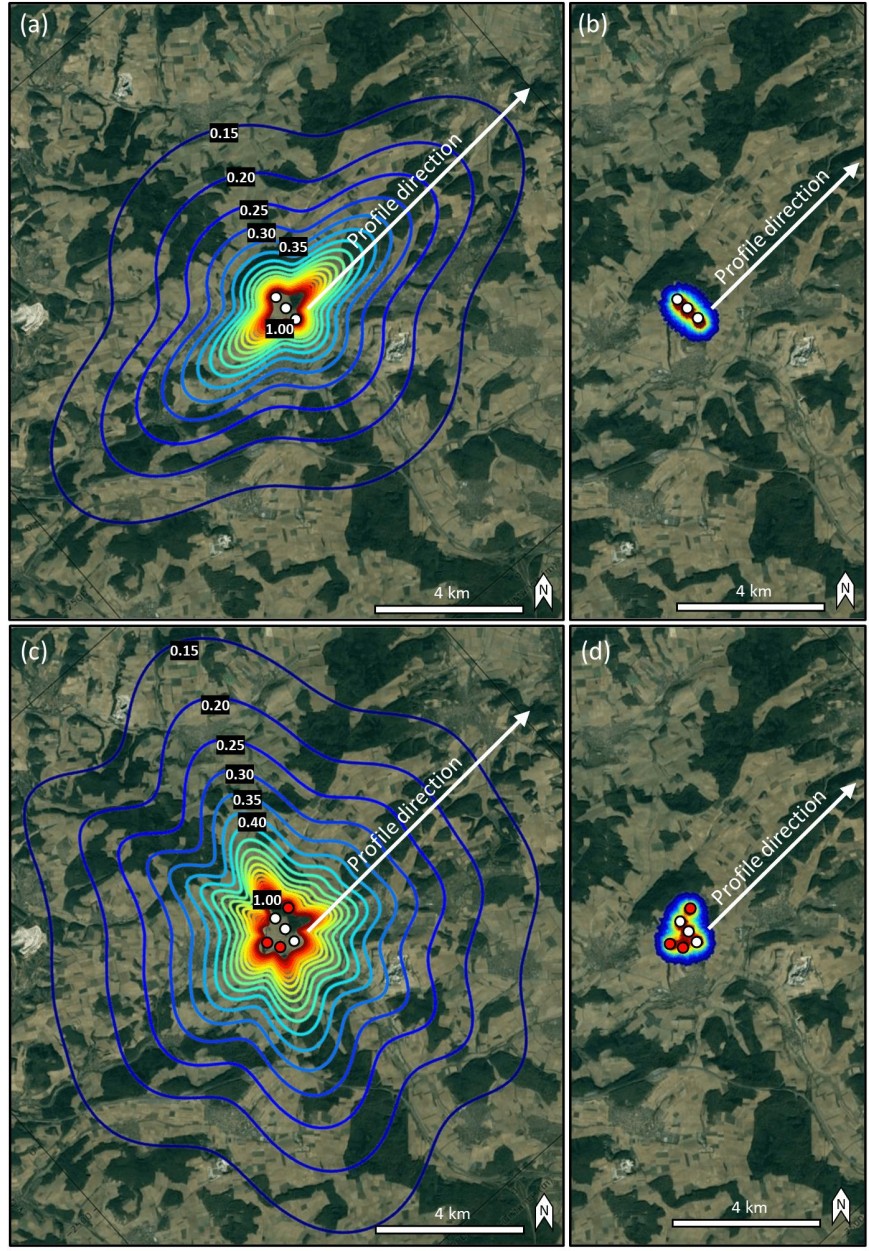

**Figure 14:** Estimated seismic radiation pattern (red=high amplitudes) of (a) the Uettingen wind farm (white dots) for 1.14 Hz and (b) 7.6 Hz. Three arbitrary WTs (red dots) are added to the existing WF and affect the radiation for (c) 1.14 Hz and (d) 7.6 Hz. Model parameters are $c_S = 960$ m/s, $Q_S = 40$ in (a) and (c), whereas $c_S = 540$ m/s, $Q_S = 16$ in (b) and (d). Calculations are based on 500 averaged wave fields using random source signal phases. Contour lines show amplitude decay factor from 1 to 0.15. Maps data: © Google 2021.





## 6 Conclusions

We recorded the seismic signals emitted from a 3-turbine WF in Uettingen, Bavaria over a period of 6 months and analyzed
the spectral characteristics and spatial amplitude decays. During "full power" operation mode of the WTs we identify 7
prominent spectral peaks in the frequency range from 1.14 Hz to 7.6 Hz. The attenuation of the peak amplitudes with respect
to the WT distances can be described by a power law with exponent $b$. We find that the calculated $b$-values increase linearly
with increasing peak frequency and range between 0.39 and 3.93. Due to the relatively long observation period the calculated
values provide a stable basis for the analytical simulation of the emitted wave field.

An analytical approach was developed to model the seismic radiation of the WF. From measurements we observe that WTs
are not vibrating in phase and that the phase differences vary randomly over time. Furthermore, the results of the simulation
show a strong influence of phase differences between single WT source signals on the radiation pattern and hence on the spatial
amplitude decays. We applied a phase-shift-elimination-method (PSE-method) to eliminate this effect with the aim to derive
a representative seismic wave field. Modeling results were compared to the observed frequency-dependent amplitude decays

to derive model parameters ($Q_S$ and $c_S$) for a two-layer model which provides information about the local geology. Concerning
the modeling of WT-induced seismic signals, we can show that the signal phases of multiple source signals (multiple WTs)
have significant influence on the seismic radiation of the WFs. This effect should be carefully considered when selecting
suitable source signals to avoid misleading simulation results.





# Appendix

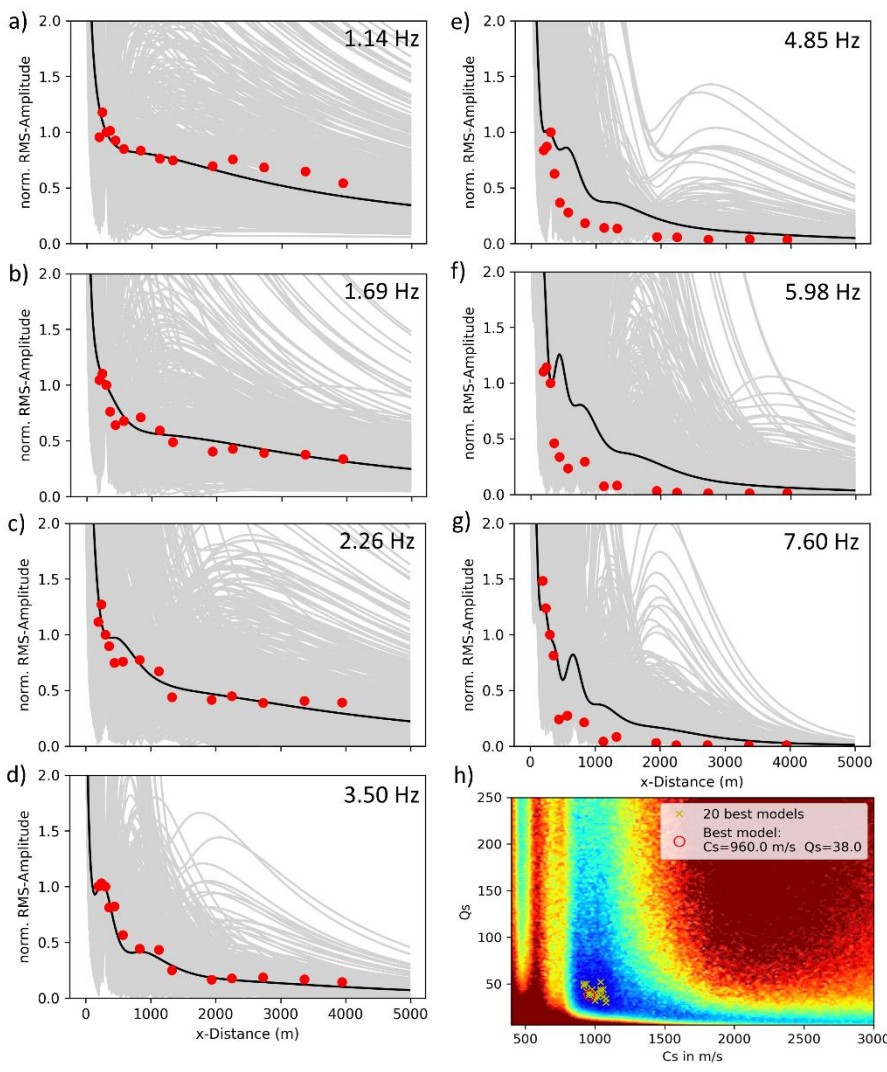


**Figure A1:** (a)-(g) Comparison of calculated and observed data (red dots) for seven signal frequencies assuming a homogeneous underground model. (h) Best model parameters are $c_S = 960$ m/s and $Q = 38$ to fit all data simultaneously.




**Code an data availability.** The code and data used in this research are currently restricted.

**Author contributions.** FL and ML performed the field work and data analysis. FL set up the modeling approach and performed
model calculations. GR participated in data interpretation, model development and supervised the article outline. HD and GR
initiated the project and provided the computational framework. FL, ML, GR and HD edited the article.

**Competing interests.** The authors declare that they have no conflict of interest.

**Acknowledgements.** We would like to thank ESWE Versorgungs AG for providing access to and operational data from the
wind farm facilities in Uettingen and especially U. Schneider for his support in initiating the project.  We appreciate the support
of the mayors of Uettingen, Greußenheim, Remlingen and Leinach as well as municipal administrations. Special thanks to W.
Reinhart, J. Palm and A. Costa for their help during field work and station maintenance and to G. Schmidt (ESWE), our contact
person regarding technical matters of the WF Uettingen. This study is part of the project KWISS which is funded by the
*German Federal Ministry for Economic Affairs and Energy* (FKZ 0324360) and *ESWE Innovations- und Klimaschutzfonds*.

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
