# Peer review of "Seismic radiation from wind turbines: observations and analytical modeling of frequency-dependent amplitude decays"

_Solid Earth, 2021_

## Author Comment (AC1)

**Comment on se-2021-21 by Anonymous Referee #1**

We sincerely thank the referee for the valuable comments which will help to improve the paper. Please find a point-by-point response to the comments. Our responses are in red. Actual changes to the text are marked as *italic* font.

On behalf of all authors, Yours sincerely,

Fabian Limberger

**Comments by Anonymous Referee #1:**

The work is quite interesting, it is also well written (with some small exceptions, as page10).

The description on page 10 will be improved in the revised manuscript. It should be clearer after these changes.

The authors propose a way to model the seismic motion produced by the operation of wind turbines. They show the importance of phase shift due to the presence of more than one wind turbine, and propose a method to remove it. However, the main problem in the manuscript is the lack of discussion of the effect that more wind farms would have in both: the observed and the modeled motion.

Yes, the focus of this paper is on the "phase-shift-elimination" method which can be used to estimate the amplitude field around a simulated windfarm. At the moment we cannot reliably estimate the impact of wind parks A and B but there are good reasons to assume that, for the considered distance range (max 4 km), the measured amplitudes are dominated by the wind turbines in Uettingen. Lower-frequency signals from other windfarms may have a stronger impact and this is partially discussed already. We will improve on the discussion during the revision. However, the detailed analysis of the contribution of neighboring windfarms on the measured amplitudes is beyond the scope of this paper. To consider their effects additional field data from an optimized station layout are required. We are currently preparing for field measurements to estimate this influence, but the results will not affect the general idea of the presented method.

The spectral analysis and their observations are consistent with previous works. However, in their results they mention "The remaining (sharp) peaks show no systematic dependence which is an indication that their origin is not related to the WTs." To what could it be related? Could these peaks be related to the other two existent windfarms? Which are in some cases, closer than the Uettingen WF to the recording stations (i.e., stations 4 km away).

The sharp peaks show no systematic distance dependence. Neither to the Uettingen wind farm nor to the neighboring windfarms. The amplitudes of these minor peaks show no correlation with the rotation rate of the wind turbines, or the wind speed. This indicates that the source which is causing these sharp peaks is probably not related to the operation of the wind farms. Therefore, these peaks were not analyzed in detail because amplitudes are small compared to the analyzed major peaks.

When the authors analyze the amplitude decay with the distance between 300 and 4000 m there are some stations with discrepancies with the fitting power law, which the authors explain as an effect of the near field for stations ~ 300 m away from the WF and as local anthropogenic noise for the stations at more than 3 km distance (which actually had been removed). So again the question would be: could

these effects be due to the other two wind farms? For the farther stations, which would be the role of the eastern wind farm with six wind turbines?

With the averaging process we can remove only transient signals. Towards the end of the profile (F15 - F19) we are relatively close to the northeastern windfarm. However, the stations are also close to a heavily used highspeed railroad line. Both could explain the elevated PSD amplitudes which we observe in this region. This will be analyzed by future measurements. So far, we observe at these stations a general elevated noise level rather than an increase of single PSD peaks. This may be an indication, that the source is not the neighboring windfarm.

On the discussion, the authors mention shortly the effect that short measurements (shorter as 6,5 weeks) can have on the estimation of b values, because of the presence of transients and earthquake events. On section 2.1 the authors mention they removed the local transients, but they don't explain how they managed with earthquake events. The authors should clarify if they removed these signals or how they managed with them.

Earthquake signals were not removed explicitly before averaging the PSD spectra. Keeping in mind that we used several thousands of single PSD spectra, we think that the influence of the (relatively few) earthquakes on the average amplitudes should be negligible. Furthermore, we removed 75% of the largest amplitudes (outliers) before calculating the average. This also reduces the influence of transient signals on the average amplitude calculation.

The discussion of the authors is good and complete, and they focus on the problems they solve. The authors show the important role that three aleatory wind turbines would have in the motion, but the role of the nearby wind farms (with even six turbines) is just shortly mentioned. Please discuss in more detail.

The discussed comments on the additional wind farms will be included in the discussion of the revised article in more detail. We will add the following text to the discussion in line 346 after "…targeted three WTs in Uettingen (Fig. 1)":

*This could lead to an overestimation of the signal amplitudes, especially in the far field of the WF Uettingen. However, since we observe peaks at identical frequency in the near and far field of the WF, it is reasonable to assign these signals mainly to the wave field produced by the WTs in Uettingen. Signals from various WFs can generally be distinguish using e.g. mitigation analysis (Friedrich et. al., 2019). However, the detailed analysis of the effect of additional WFs around Uettingen is beyond the scope of this study but should be considered in future analysis.*

Would it be possible to identify which signals are really coming from the Uettingen WF and which from the other wind farms, in order to identify the origin and obtain an even better model?

In ongoing analyses, we are working on the cross correlation of signals measured along the profile to identify signals that are emitted by the WTs in Uettingen. Generally, there are other approaches to distinguish signals from various wind farms (e.g. Friedrich et. al., 2019). But this is beyond the scope of our article, which focuses on the aspects of using an analytical approach to model the radiation patterns of a wind farm.

Figures 12 and 13 are discussed in the text before figures 10 and 11, it would be better to change the order. Figures 10 and 11 need a color scale for the modeled radiation patterns. The caption of figure 13 should be improved.

We agree with the comment on the order of the figures 10-13 and will change the order in the revised manuscript. Furthermore, we will add the color scale to figure 10 and 11.

The caption of figure 13 is improved to:

*Figure 13: Two-layer model derived by fitting observed and modeled amplitude decays. The best model parameters (cS and QS) for the two layers are found by performing a grid search to optimize the fitting of amplitude decays of signals < 4 Hz and > 4 Hz separately. The depths of the layer interfaces are obtained by assuming a penetration depth of surface waves of λ/3. The transition between layer 1 and layer 2 is somewhat obscure due to the lack of amplitude decays related to signals between frequencies of 3.5 and 4.85 Hz.*

---

## Author Comment (AC2)

**Comment on se-2021-21 by Joachim Ritter Referee #2**

We sincerely thank Joachim Ritter for the valuable comments which will help to improve the paper. Please find a point-by-point response to the comments. Our responses are in red. Actual changes to the text are marked as *italic* font.

On behalf of all authors, Yours sincerely,

Fabian Limberger

**Comments by Joachim Ritter Referee #2:**

The manuscript on 'Seismic radiation from wind turbines: observations and analytical modeling of frequency-dependent amplitude decays' is an important contribution to better understand and predict seismic emissions from wind turbines. Measurement results from a well-chosen experiment are presented together with a new approach to model emissions from several wind turbines such as typical wind farm installations.

The two main results are clearly outlined: attenuation factors for a long-term measurement (6 month) and the influence of phase shifts from multiple sources on the emission amplitudes. However, I recommend a revision before publication:

A description of the **geology / underground** is completely missing (e.g. after line 68). This information is important to understand seismic velocities and quality factors which depend on the physical rock properties.

We certainly agree and will add this information to the revised paper:

*The local near surface geology is defined by Triassic sedimentary rocks. Beneath a thin soil layer limestones of the Muschelkalk are situated over clastic sediments of the Buntsandstein, mainly terrestrial quartzite, sandstone, and claystone layers. Geologic cross sections suggest that the Lower Muschelkalk under the topographic surface reaches a thickness of up to several tens of meters (Bayerisches Geologisches Landesamt, 1978). However, at some seismic stations the Muschelkalk/Buntsandstein boundary is only a few meters below the surface. In topographic depressions the Muschelkalk can be completely missing, i.e. thin quartenary soft sediments directly cover Upper Buntsandstein rocks.*

*Bayerisches Geologisches Landesamt, 1978, Geologische Karte von Bayern 1:25.000, Blatt 6124 Remlingen, München.*

The influence **of wind parks A and B** (Fig. 1) must be explained in more details. Are there large wind turbines which may affect the measurements ? What happens if they radiate emissions in phase ?

This is an important question. At the moment we cannot reliably estimate the impact of wind parks A and B. Our observations show that for the considered distance range (max 4 km) the measured amplitudes at frequencies about > 4 Hz are dominated by the wind turbines in Uettingen. For lower frequencies, there could be some influence which we will quantify further in a forthcoming study.  In the current paper we focus on the phase shift elimination method and its potential applications. To estimate the influence of the neighboring wind parks we are planning additional measurements in the future.

What is exactly meant by using the **25% quantile** (line 81). Does this mean you exclude 75% of the data ? How sensitive is this selection ? Would it make a difference to use the 20% or 40% quantile ? Do you exclude time windows with earthquakes waves ? This should be clarified in the manuscript.

Yes, it means that we exclude 75% of the largest PSD amplitudes, which we consider as outliers. Since we have a very high number of time windows (and hence PSDs), we can remove such a high amount of data to derive very robust and clear averaged PSD spectra. Time windows with transient signals e.q. from earthquakes or human traffic have not been excluded.

For clarification we suggest to include the following figure and explanations in the manuscript:

[Figure]

Figure R1: Influence of different percentiles on the calculated average PSD functions:

The figure shows the single PSD spectra for station F01 as **grey lines** in the background. These spectra were calculated from 10-minute windows and cover a time range of seven months (Sep 2019 to Mar 2020). In total, there are 9855 individual spectra that have been registered during the "full power" status of the wind turbines (11-12 rpm).

The **black line** represents the average PSD of all single spectra (100% percentile). There are numerous peaks, however, they are partly masked by the general high amplitude level, especially at frequencies below 3 Hz. This "noise" is probably caused by transient signals like earthquakes or anthropogenic signals, which are not explicitly identified and excluded. It can be minimized by removing the largest outliers before averaging the single PSDs.

The **colored lines** represent average PSD spectra calculated from different percentiles of the single PSD functions (blue = 75%, green = 50%, and red = 25%). Now the different PSD peaks have been improved and can be identified more safely.

Although the effects of different percentiles are relatively small, we decided to use the 25% percentile to derive the amplitude decay of the WT signals (i.e. 75% of the largest amplitudes have been removed before averaging). We think that this gives a reliable and conservative estimate of the spectral WT amplitudes with a minimized influence of interfering transient signals. Due to the long observation period, there is still enough data left (2463 single spectra) to calculate robust average PSDs.

The peaks at relatively high frequencies of 6.0 Hz and 7.6 Hz are the highest ones. How can this be explained ? Which operational modes of the wind turbines are these ? Please explain.

We had discussions with engineers for the structural monitoring of wind turbines. We could not clarify the origin of these peaks so far. However, maximum peaks at ~6Hz are also measured by Lerbs et. al. (2020) and Neuffer et. al. (2019), for example. These peaks are very prominent only in the near field and exhibit a rapid amplitude reduction with distance, as our observations show. In terms of studying the vibrational modes of a wind turbine, these signals are potentially very interesting. In terms of studying the wave propagation along the profile to the far field, these signals play a secondary role. Therefore, we do not focus on assigning operational modes to these high frequencies.

Section 2.2 on amplitude decay: this is on **PSD** amplitude decay as described in lines 104-106. To makes this clear, **PSD** amplitude decay should be written in lines 106, 109 and 121.

In general, our paper deals with PSD amplitudes and their decay. Therefore, confusion with time domain amplitudes should be avoided. However, we can add the prefix "PSD" at the appropriate places in the text as proposed by the reviewer.

It would be helpful to include the amplitude decay for **waves in the time domain**. Therefore, a conversion should be applied (factor 0.5). Then the resulting b-values can be better compared with typical wave properties, e.g. 0.5 for geometrical spreading of surface waves.

The primary results of our analyses are the b-values for PSD amplitudes. We therefore prefer to retain these values in figures and discussion. For a better comparison we will add a table which lists the frequency dependent b-values (PSD) together with the corresponding b-values for time domain amplitudes (simply apply factor 0.5).

| $f$ (Hz) | $b$ (PSD) | $b$ (AMP) | $b$ (RMS) |
|---|---|---|---|
| 1.13 | 0.39 | 0.20 | 0.13 |
| 1.7 | 0.73 | 0.37 | 0.24 |
| 2.3 | 0.76 | 0.38 | 0.48 |
| 3.5 | 1.57 | 0.79 | 0.81 |
| 4.8 | 2.40 | 1.20 | 1.10 |
| 6.0 | 3.39 | 1.70 | 1.54 |
| 7.6 | 3.93 | 1.97 | 1.99 |

Table R1: Listing of the measured b-values for the PSD-amplitude decay, b (PSD), and corresponding b-values of the signal-amplitude decay, b (AMP). The latter was derived from b (PSD) by application of factor 0.5.

For the referee's information we have included *b*-values measured in the time domain by averaging RMS amplitudes (root-mean-square) of 10-minute time segments of the WT signals, *b* (RMS), in green. These *b*-values correspond fairly well with *b* (AMP) and hence confirm the measured PSD-amplitude decay, *b* (PSD). However, we do not intend to present these measurements and results in our manuscript, because this would exaggerate the section on *b*-value calculation, whereas the focus of the paper should be on the modeling approach.

In addition: in **Fig. 5, the b-values from Lerbs et al. correspond to wave amplitude decays, not PSD decays !** This should be checked also for the values in Neuffer et al. (2019).

Regarding the paper of **Lerbs et al.2020:** We refer to Figure 4 in that paper. Here, labels and figure captions clearly indicate PSD values. Similar, Fig. 5 of Lerbs et al. 2020 shows PSD spectra, as stated in the figure caption.

With respect to **Neuffer et al. 2019:** We refer to Figure 10 in that paper. Based on the captions, these are clearly PSD amplitudes. The peculiarity here is that the authors analyze not the PSD amplitudes at high windspeeds (corresponding to full power status) but the differential PSD amplitudes, which means the differences between PSD amplitudes at high windspeeds (> 12 m/s) and low windspeeds (< 1 m/s)

Fig. 6 nicely shows the effect of the random phase shifts. Can your observations explain the observation by Neuffer et al. (2019) that the emissions amplitudes increase with the square root of the number of wind turbines?

Since we analyze signals that are emitted from only 3 WTs, it is difficult to show the increase of amplitudes. We refer to Fig. 12 in Neuffer et. al. (2019). Here, the first three data points could also be fitted by a linear curve instead of an N^0.5 fitting. The nicely shown N^0.5 relation appears much clearer when 5 or more WTs are considered. The experiment of Neuffer et. al. could work as a benchmark to check this relation with our simulation approach. However, this is beyond the scope of our current paper, but could be considered in a future study.

Line 165: Quality factor Q: You do not mention seismic scattering. Especially for high frequencies this may contribute to the wave damping.

Yes, scattering of waves generally reduces the amplitude. We agree that there may be an effect on the damping of the amplitudes. However, in the area of Uettingen we do not expect very strong effects of topography or fractured zones since the topography is relatively smooth and large damaging zones (or faults) are missing. Within our current study, we decided to include intrinsic attenuation only, which may result in a slight overestimation of the intrinsic damping.

Equation (3): A constant amplitude A is used. I understand that this is a reasonable start for modelling. However, it should be mentioned that wind turbines emit timely and azimuthally varying signal amplitudes (e.g. Lerbs et al., 2020: Fig. 6-8). This will modify the results below as well as should be considered in real cases.

That is correct and we can clearly observe time variant amplitudes and particle motions at the sources. Regarding long time periods, we find that a "representative" source amplitude can be assumed for the same type of turbines. However, this is clearly not true if short time periods are considered. We will describe these differences in more detail in the revised version.

Section 3.2: It would be helpful to add some sentences on the wave pattern off the profile and add comments for a non-uniform signal amplitude A (eq. 3).

Our approach assumes that wind turbines of the same type emit comparable signals. This is statistically expectable regarding long time periods, which we are analyzing. If a wind farm has

different WT types, the individual source amplitudes should be adjusted, which generally can be done with the presented approach, if necessary.

The wave pattern off the profile is quite symmetrical since the turbines are positioned in a clear geometry (in a line with similar distances between the turbines). We will also add a comment on this in the revised version of the paper.

4 Results: The Q value for low frequencies (40) is quite low. Does this fit with the physical properties of the rocks in the subsurface at >100 m depth ? I guess this could be a solid and compact limestone.

Referring to the local geology (see response to an earlier comment above), we expect mainly rocks of the Buntsandstein below ca. 100m depth. Generally, rocks of the Buntsandstein has lower Q values than limestones. Furthermore, the Q value in this study corresponds to the attenuation of the S-wave. Q values for P-waves are higher in most cases. Referring to laboratory experiments of e.g. Johnston and Toksöz (1980)\*, Q values for the S-wave in dry limestone are around 40-80 (at confining pressures of 50 to 1000 bars). However, the frequency range given in their paper is between 0.1 and 1.0 MHz and we are aware that these results can not directly be compared to the frequency range of WT signals (1-10Hz). However, it is known that the Q value itself tends to decrease slightly with decreasing frequency. Based on these points, we would conclude that our findings are within a reasonable range of values to describe near-surface wave propagation and damping. But we agree that modeled Q and $V_S$ values might be biased by the influence of wind parks A and B, which we cannot fully quantify up to now.

\* Johnston, D. H., & Toksöz, M. N. (1980). Ultrasonic P and S wave attenuation in dry and saturated rocks under pressure. Journal of Geophysical Research, 85(B2), 925. doi:10.1029/jb085ib02p00925

Generally, I am missing a comparison of the vs and Q values with the actual rocks along the profile.

We agree, see our remarks given in response to an earlier comment above.

Results and Fig. 13: What requires a third layer (half space below the second layer) ? Why don't you use a one-layer model with a half-space below ?

We used a one-layer model (called "layer 1") with a half-space below (called "layer 2"), see fig. 13. However, since we only have attenuation information about signals down to a frequency of 1.14 Hz, we are limited concerning the conversion of wavelength to depth information. Therefore, we can only derive values ($V_S$ and Q) to a (estimated) depth of about 280m and we cannot give information about the properties of deeper layers. Our presentation in Fig. 13 may be not clear and we will adjust its description in the revised manuscript.

Fig. 10 and 11: The gray model curves on the left do not indicate a preference for the average model (black) line. Also the scatter around the black curve is unclear. Better use a colour / gray scale which indicates the actual distribution density of the models. A colour scale for the right part is missing, so it cannot be understood in its current version.

So far, the gray curves only indicate the variation of possible decay curves. On the one hand, we agree that it would be generally interesting to understand the true distribution density of the models. On the other hand, the observational data (PSDs) is highly averaged. The thousands of PSDs that are averaged contain a very high number of different (random) phase constellations, caused by the time variant signal phases of the sources. To compare the observed averaged amplitudes with the modeled data, we therefore prefer to keep using the average of the models instead of the density distribution.

However, Fig. R2 shows an example, as the referee suggested.

[Figure]

Figure R2: Example of two modified figures including the distribution density of the models in the left part, as the referee suggested. Top: f=1.14Hz, bottom: f=3.5Hz. The blue line is the average of all simulated models. The gray color indicates the distribution density of the models. The red dots are the observed amplitudes.

The color scale for the right part will be added, as suggested by the reviewer.

Line 309: use b-values for waves in the time domain; if b is still smaller than 0.5 please explain why the wave amplitude decay is smaller than geometrical spreading

An explanation for this is the interference of the three wave fields from the three WTs. Although the decay of signals from a single source would not be lower than the geometrical spreading attenuation, the superposition of wave fields could increase the amplitudes along the profile. We will add appropriate comments in the revised manuscript.

Lines 311-315: make sure that the b-values of the other studies are comparable to your values (PSD decay, wave decay, …)

As mentioned above, the b-values of the cited authors refer to PSD-decay. Only the results presented by Flores Estrella et al. (2017) seem to relate to spectral amplitudes rather than PSD amplitudes. However, there is some uncertainty on that. Therefore, we do not discuss their b-values in our manuscript.

The discussion should include a paragraph comparing the vs and Q with the rock properties at depth.

See comments given above.

Others:

Line 35: were able **to** distinguish

We will correct this.

Line 44: Zieger **et al.**, 2020

We will correct this.

Line 61: rpm (**rotations** per minute)

We agree that "rotations per minute" is better to understand, even if both (revolutions and rotations) is common to use.

Line 84/85: include Hz after the numerical values

We will add Hz after the values.

Line 234: Explain why you use the square root for the conversion ? Do you want amplitudes in the frequency or time domain ?

Measurements are in PSD units, modeling results are in RMS amplitude units. With the conversion we want to make them comparable. In our study, we do compare relative amplitudes within one single peak frequency, however, we never compare amplitudes between different frequencies. Therefore, a conversion of the "spectral" PSD-values to RMS-amplitudes in the time domain can be performed using the square root of the PSD values without losing the correct relation between the amplitudes of same frequencies measured at the stations. A complete conversion to derive correct absolute RMS-amplitudes from PSD-amplitudes would necessarily include a scaling factor based on the frequency range we are looking at. However, this is not necessary using relative amplitudes for one single frequency.

Line 238: 6000 **m**

We will correct this.

Lines 265: not "until" but "**down to**"

We will correct this.

Line 266: Fig. 13 appears in the text before Fig. 10-12 – please re-sort.

We agree that this is not intuitive. The Figures will be re-organized.

Figure 1: Würzburg must be shown (it is cited in the text)

To include Würzburg, the map scaling must be changed so that details of the station setup would be lost. We therefore decided to indicate only the direction to this city.

[Figure]

(Modified Fig. 1, indicating the direction to the city of Würzburg)

Fig. 12: the black x (better in white) and the red circle (better in orange) can hardly be seen, increase the contrast

We will do this, but instead of orange circles we choose magenta. The contrast is even higher then.

**Citation**: https://doi.org/10.5194/se-2021-21-RC2

---

## Author Response (AR2)

**Comment *(minor revision)* on se-2021-21 by Joachim Ritter Referee #2**

We sincerely thank Joachim Ritter for the valuable comments which will help to improve the paper. Please find a point-by-point response to the comments. Our responses are in red.

On behalf of all authors, Yours sincerely,

Fabian Limberger

**Comments *(minor revision)* by Joachim Ritter Referee #2:**

Dear authors,

the revision of the manuscript is well done in my opinion. Before publication I only recommend one very minor point:

On page 10, line 177 you mention that `Previous research suggests that mainly vertically polarized Rayleigh waves are emitted …`. Looking at two recent papers, it seems that Love waves are emitted in the same intensity as Rayleigh waves.

1) The polarization diagrams in Lerbs et al. (2020) Fig. 8 indicate a clear polarization perpendicular to the line between the wind turbine and the recording station. This is best explained by Love waves. In fact, it seems that Rayleigh waves and Love wave alternate.

This result coincides well to a

2) A new paper by Neuffer et al. (2021, Journal of Seismology, 25, pages 825–844). They also find strong indications for Love wave emission from polarization analysis on the foundation of a wind turbine.

Perhaps these indications for Love waves can be added on page 10.

Otherwise the paper can be published in its revised version.

Best regards, Joachim Ritter

We agree with these suggestions. We added a sentence on page 10 and included the article of Neuffer et al. (2021) in the reference list.